# Cost-Effectiveness Analysis of Cell Versus Egg-Based Seasonal Influenza Vaccination in Children and Adults in Argentina

**DOI:** 10.3390/vaccines10101627

**Published:** 2022-09-28

**Authors:** Analía Urueña, Paula Micone, María Cecilia Magneres, Ian McGovern, Joaquin Mould-Quevedo, Túlio Tadeu Rocha Sarmento, Norberto Giglio

**Affiliations:** 1Centre for the Study of Prevention and Control of Transmissible Diseases(CEPyCET), ISalud University, Buenos Aires C1095AAS, Argentina; 2Gynecology Department, Hospital Carlos G Durand, Buenos Aires C1095AAS, Argentina; 3Seqirus S.A, Medical Affairs, Buenos Aires C1095AAS, Argentina; 4Seqirus USA Inc., Medical Affairs, Summit, NJ 07901, USA; 5Cerner Enviza, São Paulo 04794, Brazil; 6Epidemiology Department, Hospital de Niños Ricardo Gutiérrez, Buenos Aires C1095AAS, Argentina

**Keywords:** influenza vaccination, cell-based vaccine, cost-effectiveness, Argentina, egg adaptation

## Abstract

Background: Quadrivalent cell-based influenza vaccines (QIVc) avoid egg-adaptive mutations and can be more effective than traditional quadrivalent egg-based influenza vaccines (QIVe). This analysis compared the cost-effectiveness of QIVc and QIVe in Argentinian populations < 65 years old from the payer and societal perspectives. Methods: A static decision tree model compared the costs and health benefits of vaccination with QIVc vs. QIVe using a one-year time horizon. The relative vaccine effectiveness of QIVc vs. QIVe was assumed to be 8.1% for children and 11.4% for adults. An alternative high egg-adaptation scenario was also assessed. Model inputs were sourced from Argentina or the international literature. Deterministic and probabilistic sensitivity analyses were performed. Results: Compared to QIVe, QIVc would prevent 17,857 general practitioner visits, 2418 complications, 816 hospitalizations, and 12 deaths per year. From the payers’ perspective, the incremental cost-effectiveness ratio per quality-adjusted life years gained was USD12,214 in the base case and USD2311 in the high egg-adaptation scenario. QIVc was cost-saving from the societal perspective in both scenarios. Conclusions: QIVc in Argentina would be cost-effective relative to QIVe. The potential health benefits and savings would be even higher in high egg-adaptation seasons.

## 1. Introduction

Seasonal influenza occurs in epidemics and affects people of all ages worldwide, with approximately 3–5 million cases of severe illness and 250,000 to 500,000 influenza-related deaths yearly [1]. In addition to common symptoms such as fever, myalgia, and rhinitis, seasonal influenza can lead to more severe consequences such as pneumonia [2] and heart diseases [3]. In Argentina, the number of yearly cases peaks from May to October [4,5], resulting in a significant disease burden, healthcare costs, and productivity loss.

At present, vaccination remains the most effective intervention to prevent seasonal influenza [1]. Vaccination and influenza treatment reduce disease burden by lowering morbidity, mortality, and direct and indirect cost in terms of healthcare resource utilization and productivity lost [6]. Since the 1980s, two distinct lineages of the B strain virus (B/Victoria and B/Yamagata) have circulated in both hemispheres. Quadrivalent influenza vaccines (QIVs) [7] include influenza A and B strains and incorporate both strain B lineages [8], thereby offering better immune coverage than trivalent influenza vaccines in terms of B mismatch [9]. Since 2013, QIVs have been recommended by the World Health Organization (WHO) for use in each influenza season [10].

For more than 70 years, the traditional methods for making millions of influenza vaccines have been egg-based techniques. Egg-based vaccines can result in egg-adaptive mutations. The abundance of avian-like receptors on the chorioallantoic membrane usually results in the selection of variants that increase binding to avian-like receptors and reduce binding to human-like receptors [11]. As a result, egg adaptation may lead to antigenic mismatch, loss of antigenicity, and consequent reduction in vaccine effectiveness (VE) [11,12]. While it is not possible to predict which seasons will be impacted by egg adaptation, a historical analysis suggested that egg adaptation occurs in more than half of the influenza seasons [13].

Egg adaptations are avoided with more recent production techniques, including the cell-based method, where the selected influenza strain is cultured in mammalian cell lines instead of eggs [14]. The development of vaccines using cell-based methods prevents egg-adaptive structural mutations to the selected strain and loss of antigenicity of the seasonal influenza vaccine. The difference between selected strains and vaccine strains is minimized with cell-based quadrivalent influenza vaccines (QIVc). The cell-based influenza vaccine developed by Seqirus (Flucelvax^®^ Quad) was shown to be more effective in preventing seasonal influenza compared to QIVe across three consecutive influenza seasons in the US [12,13,14,15]. In addition, a recent study on expert consensus found that the effectiveness of influenza vaccines increases as much as 9% if processes leading to egg adaptations are avoided [15].

The cost-effectiveness of QIVc has been reported for the United States (US) [16,17], Spain [18], Brazil [19], the United Kingdom (UK) [20], and Germany [21]. In the UK, Kohli et al. reported the cost-effectiveness of extending QIVc vaccination regimen from at-risk to low-risk adults between 50 and 64 years of age [20]. QIVc was found to be a cost-effective alternative to QIVe in a seasonal influenza analysis using an economic model on high-risk subjects aged between 9 and 64 years in Spain [18]. In Brazil, replacing the trivalent vaccine with QIVc in the Brazilian National Immunization Program proved to be cost-effective [19]. Corroborating this evidence, reports from real-world data from the US during the 2017–2020 influenza season demonstrated that QIVc was significantly more effective in preventing influenza-related hospitalizations and emergency room visits, all-cause hospitalizations, and any event-associated hospitalizations and emergency visits compared to QIVe. QIVc in this period was also associated with significantly lower all-cause costs [17,22,23].

In 2011, the Argentina National Immunization Program implemented annual influenza vaccination for high-risk groups including children between 6 months to 2 years, adults > 65 years, healthcare workers, pregnant women, and subjects with comorbid conditions and chronic diseases between 2 and 64 years [24]. Since then, egg-based influenza vaccines recommended by the WHO have been used, and from 2020 onwards, adjuvanted influenza vaccinations have been recommended for individuals over 65 years. The aim of the present study was to assess the cost-effectiveness of QIVc vs. QIVe in the recommended target populations (children < 2 years and children and adults < 65 years) in Argentina from the payer and societal perspectives.

## 2. Materials and Methods

A decision-tree model was developed using Microsoft Excel^®^ (Microsoft Corporation, Redmond, WA, USA) to address the study’s objectives. Given the clinical and epidemiological circumstances of influenza, a one-year time horizon was adopted. The methodology was similar to previously published cost-effectiveness analyses of influenza vaccination in South America [4,19,25,26].

### 2.1. Intervention and Comparators

Clinical outcomes and healthcare resource utilization and costs due to influenza cases were assessed for scenarios under QIVc and QIVe vaccination in high-risk Argentinian populations.

### 2.2. Target Population

The most recent estimates of Argentinian population were used [27] with the following age groups: (I) 6 months and 24 months old; (II) 2–4 years old; (III) 5–14 years old; and (IV) 15–64 years old. For the groups between 2 and 64 years old, only the population with risk factors was considered in the model, with estimates provided by the National Immunization Program (NIP) and based on the National Risk Factors Survey [28]. The vaccination coverage rates for each age group were provided by the NIP and were used in the model for both vaccinations (Table 1) [29].

### 2.3. Model Structure

To properly address all relevant characteristics of influenza clinical evolution, a comprehensive set of outcomes was adopted: number of influenza-related General Practitioner (GP) visits; number of influenza-related complications; number of influenza-related hospitalizations; number of influenza-related deaths; and Quality-Adjusted Life Years (QALY) gained. The costs were measured for 2021 and are presented in United States Dollars (USD), with an exchange rate of 1 USD = 97.25 ARS [30].

At model entry, the subjects were assigned to be immunized with either QIVc or QIVe (Figure 1). After receiving the vaccines, the individuals could be exposed to influenza viruses (both A and B strains) and were infected or not; 81.8% of all circulating influenza viruses were A strains, and 18.2% were B strains. These proportions were estimated by the authors in a previous report from an average circulation during the 2014 to 2019 seasons and were based on information provided by the National Epidemiology Directory [4]. After virus exposure, the chance of an individual becoming infected was defined by influenza incidence (from the National Epidemiology Directory [4]) and then adjusted for the attack rates for each strain [31].

The absolute effectiveness of QIVe used in this model was obtained from a systematic literature review and meta-analysis [32]. The relative vaccine effectiveness (rVE) of QIVc vs. QIVe was estimated using a non-systematic structured review to identify publications reporting relative vaccine effectiveness estimates for interventions and comparators for prevention of influenza-related medical encounters (IRME) in any clinical setting. Four publications corresponding to three studies were identified (Imran et al. (2022) reported the results for children/adolescents and adults in separate publications), and their reported effect estimates were pooled using DerSimonian and Laird random effects meta-analysis (separately for individuals 4–17 and 18–64 years old) [33,34,35]. Based on the results of the meta-analysis, the rVE of QIVc vs. QIVe for prevention of IRMEs was assumed to be 8.1% (95% confidence interval (95% CI) 0.1% to 15.4%) for children (6 months–14 years old) (Figure 2) and 11.4% (95% CI 5.8% to 16.7%) for adolescents and adults (15–64 years old) (Figure 3). Due the lack of data available comparing QIVc and QIVe for children under 4 years old, it was assumed that the rVE for children under 4 years old would be equivalent to the one obtained for children 4–17 years old. Both the 4–17 and 18–64 meta-analyses had high heterogeneity (I^2^ > 75%), as may be expected due to season-to-season variation in the underlying epidemiological characteristics of a season (e.g., variable effects of antigenic drift and egg adaptation). The large sample sizes included in these studies and the resulting narrow confidence intervals further highlighted these differences and increased heterogeneity.

According to the model rationale, if subjects become infected, they might experience minor symptoms or develop complications that could be managed as outpatients or require hospitalization. Only hospitalized individuals could progress to death in the model (Figure 1). Although this may understate the total number of deaths, this rationale was maintained due to the existence of a universal health system in Argentina, where it is expected that most complicated cases could be clinically managed, as well as the uncertainty about the proportion of deaths without clinical management in the country. Due to the lack of local data to estimate the proportion of influenza cases that might become complicated, require hospitalization, or be fatal as a result of complications, working values were taken from the international, peer-reviewed, scientific literature [25,36].

Vaccine effectiveness impacts only the probability of becoming infected—once the individuals are infected, it is expected that they are susceptible to the clinical course explained above.

Because of the lack of disutility data for influenza cases, complications, or hospitalizations specific for Argentina, disutility values were obtained from an international, peer-reviewed study by Hollmann et al. (2013) [37]. In that study, the authors estimated the disutilities for influenza using the EQ-5D formulary, stratified by age group. These data reflected the daily disutility caused by each of these clinical events. To address this point, an average length of 7.5 days was assumed based on local reports and authors’ assumptions [38].

The economic model also includes comparisons across vaccine acquisition costs, treatment and management costs associated to influenza, complications costs, and productivity losses (Table 2). The direct medical costs for treating influenza complications were obtained from an Argentinian study [4], with the values updated to 2021 based on Consumer price index [39]. The indirect costs included productivity losses (work absenteeism) due to influenza. The latter was estimated for the 18–64 year-old group only. These individuals were assumed to be absent from work during their illness, assumed to last 7.5 days. Thus, the indirect costs for work absenteeism were estimated considering the average daily wage in Argentinian employees in 2021 [40]. This analysis also included the indirect costs due to premature deaths, in which the number of years between the subjects’ death age and their expected retirement age was multiplied by the Argentinian average yearly wage [40].

The results of this cost-effectiveness analysis will be discussed in terms of the incremental cost-effectiveness ratio (ICER) of cell-based vs. egg-based quadrivalent vaccines. The ICER represents the additional resources required to achieve one unit of a desired outcome (in this case, one QALY). The ICER assesses the performance of an intervention vs. comparator and represents how a new technology would fit into a health system. This calculation is usually performed by comparing the ICER to the willingness-to-pay threshold of this health system. In the absence of a willingness-to-pay threshold specific for Argentina (following other similar economic evaluations in Argentina), the authors used the WHO criteria for cost-effectiveness thresholds—one GDP per-capita defined a highly cost-effective strategy and three GDP per capita defined a cost-effective intervention [41].

### 2.4. High Egg-Adaptation Scenario

Seasons with more prevalent adaptative mutations and higher H3N2 circulation are expected to lead to higher VE for QIVc compared to QIVe, as observed in the 2017–2018 season [42]. In seasons such as 2017–2018, the cost-effectiveness of cell-cultured vaccination might differ from the one observed in the average scenario. To address this issue, a complementary analysis scenario was developed to address seasonality and to evaluate the benefits of cell-based vaccines in such high egg-adaptation scenarios, defined as seasons in which at least one strain of the circulating viruses had low antigenic similarities (<50%) with egg-based vaccine viruses. Such low similarity was observed during the 2017–2018 season, when H3N2 circulating strains were antigenically similar to egg-based vaccine viruses in only 40% of the samples [13]. Therefore, in the high egg-adaptation scenario, the rVE of QIVc over QIVe (using IRME as the endpoint) for the 2017–2018 season was applied: 18.8% for children and 26.8% for adults [34]. Other years were not specifically assessed since the relative effectiveness observed in such seasons was not far from the average rVE adopted in the model nor presented a high egg adaptation, one of the focused outcomes of this study.

### 2.5. Sensitivity Analysis

To estimate the uncertainties within the model results and to assess the impact of key parameters, we conducted a deterministic and a probabilistic sensitivity analysis, following international pharmacoeconomic guidelines recommendations [41]. To assess the key model parameters, a deterministic (one-way) sensitivity analysis was performed. In this analysis, the inputs of the model were varied to their lowest and highest value of 95% CI (Table 1). All the parameter models were assessed in this sensitivity analysis, although only the 10 most sensitive ones are presented in the tornado diagram and discussed in this study.

The probabilistic sensitivity analysis was conducted by varying all the model’s parameters simultaneously to a random value, according to its distribution: costs were assumed to have a gamma distribution; percentage and utility parameters were assumed to have a beta distribution, similar to a previous study [19]. In this sensitivity analysis, 10,000 Monte Carlo second-order simulations were performed.

**Table 1 vaccines-10-01627-t001:** Epidemiological inputs.

Parameter (Reference)	6–23 Months (95% CI)	2–4 Years Old (95% CI)	5–14 Years Old (95% CI)	15–64 Years Old (95% CI)
Estimated number of individuals at risk, based on 2021 Argentinean population [27]	1,123,252	252,524	848,590	3,378,426
Vaccination coverage (%) † [29]	74.6%	83.0%	49.7%	49.7%
Probability of influenza (influenza incidence) in unvaccinated subjects [4]	5.7% (5.1–6.3%)	7.5% (6.8–8.3%)	8.6% (7.7–9.5%)	10.5% (9.5–11.6%)
Probability of complication in subjects with comorbidities [43]	18.29% (16.2–19.8%)	18.29% (16.20–19.8%)	18.29% (16.20–19.80%)	12.45% (11.21–13.7%)
Probability of complication in subjects without comorbidities [25]	14.1% (12.7–15.5%)	-	-	-
Probability of hospitalization in subjects with comorbidities [25,36]	3.4% (3.0–3.7%)	3.4% (3.0–3.7%)	15.8% (14.2–17.4%)	15.8% (14.2–17.4%)
Probability of hospitalization in subjects without comorbidities [25,36]	3.4% (3.0–3.7%)	-	-	-
QIVe Absolute effectiveness against A strains [32]	59.0% (53.1–64.9%)	59.0% (53.1–64.9%)	59.0% (53.1–64.9%)	61.0% (9.5–67.1%)
QIVe Absolute effectiveness against B strains [32]	66.0% (59.4–72.6%)	66.0% (59.4–72.6%)	77.0% (69.3–84.7%)	76.0% (68.4–83.6%)
Cell-based to egg-based relative vaccine effectiveness (average season) *	8.1% (0.1–15.4%)	8.1% (0.1–15.4%)	8.1% (0.1–15.4%)	11.4% (5.8–16.7%)
Cell-based to egg-based relative vaccine effectiveness (high egg-adaptation season) [34]	18.8% (16.9–20.7%)	18.8% (16.9–20.7%)	18.8% (16.9–20.7%)	26.8% (24.1–29.5%)
Daily disutility of an influenza case † [9]	0.41	0.41	0.41	0.46
Prevalence of comorbidities [44]	11.5%	11.5%	11.5%	11.5%
Daily disutility of influenza complication † [9]	0.54	0.54	0.54	0.60
Attack rate–Influenza A † [31]	12.27%	12.27%	2.32%	2.32%
Attack rate–Influenza B † [31]	5.50%	5.50%	0.59%	0.59%
Mortality on hospitalized influenza patients with complications † [36]	1.66%	4.00%	4.00%	4.00%
Mortality on hospitalized influenza patients without complications † [36]	0.04%	0.04%	0.60%	0.60%

* Data from meta-analysis. † Assumed 10% variation. CI: Confidence Interval; QIVe: Egg-based Quadrivalent Influenza Vaccine.

**Table 2 vaccines-10-01627-t002:** Costs inputs.

Parameter (Reference)	Cost (95% CI) USD
QIVc Acquisition cost (per dose) *	7.52 (6.77–8.27)
QIVe Acquisition Cost (per dose) [44]	6.27 (5.64–6.90)
Average Hospitalization Cost (per event) [4] **	1970 (1773–2134)
Average Ambulatory influenza Costs–pediatric subjects without complications (per event) [45] **	17.56 (15.80–19.32)
Average Ambulatory influenza Costs–adult subjects without complications (per event) [46] **	51.79 (46.61–56.97)
Average Ambulatory influenza Costs–subjects with complications (per event) [46]	53.50 (48.15–58.85)
Premature death costs—indirect costs (per year) [40] **	9734 (8760–10,707)
Daily absenteeism indirect costs [40] **	40.00 (36.00–44.00)

* Data provided by Seqirus Inc. ** Values updated to 2021 based on Consumer’s price index [39].

## 3. Results

The clinical and economics comparison of QIVc and QIVe vaccination showed a clinical benefit for the cell-based vaccine, which is expected to lead to reductions in GP visits, by 17,857; complications, by 2418; hospitalizations, by 816; and deaths, by 12, in an average season (Table 3). There was also a reduction in direct and indirect costs associated with the clinical management and productivity loss of influenza cases. However, the vaccination costs were higher with QIVc than QIVe (Table 3).

The ICER was 12,214 per QALY under the payer’s perspective. Under the societal perspective, the switch of QIVe to QIVc would be cost-saving (−1,553,384 USD) and more effective (282 incremental QALYs).

The one-way sensitivity analysis showed that the vaccine acquisition costs (ICER USD 4431.6–19,997.4 for QIVc variation and USD 6119.0–18,310.0 for QIVe variation), the relative vaccine effectiveness between cell-based and egg-based vaccines (ICER USD 7696.1–22,244.1 for adult rVE variation and USD 9137.7–17,867.6 for children rVE variation), and the influenza incidence rate (ICER USD 10,999.7–13,622.1) were the most sensitive parameters in the model (Figure 4).

The probabilistic sensitivity analysis for the cost-effectiveness under the payer perspective confirmed the base case scenario, where QIVc was more effective in reducing influenza cases and complications, but also showed higher overall costs than QIVe (Figure 5)). When assessing the results under WHO willingness-to-pay threshold recommendations and using the payer’s perspective, QIVc had a 39.0% chance of being highly cost-effective and 65.1% of being cost-effective when compared to QIVe vaccination [41] (Figure 6)). QIVc would not be considered cost-effective if its relative effectiveness over QIVe was decreased as low as 3.69% for adults or −7% for children.

### High Egg-Adaptation Scenario

In this complementary analysis, in high egg-adaptation seasons, the clinical benefits were higher, leading to an ICER of USD 2311.31 per QALY. With egg adaptation, QIVc vaccination is expected to lead to reductions in GP visits, by 41,091; complications, by 5661; hospitalizations, by 740; and deaths, by 29, implying 660 incremental QALYs. Furthermore, the reduction in ICER made the chances of being cost-effective and highly cost-effective, increasing to 85.7% and 99.2%, respectively, under the payer perspective (Appendix A). QIVc vaccination, from the societal perspective in a high egg-adaptation season, was also cost-saving compared to QIVe. A one-way sensitivity analysis was also performed, in which it was possible to see that the acquisition costs of QIVc and QIVe were the most sensitive parameters (Appendix A).

## 4. Discussion

Seasonal influenza causes significant morbidity and death in Argentina’s high-risk population, which includes children and people with chronic illnesses and comorbidities. Following the influenza pandemic of 2009, Argentina’s National Immunization Program implemented annual influenza vaccination for children between six months and two years, healthcare workers, pregnant women, the elderly, and high-risk individuals. There is a measurable likelihood of altering antigenicity and loss of effectiveness in egg-based vaccines due to egg-adaptive mutations. In this study, we used an analytical decision-tree model to evaluate the cost-effectiveness of QIVc compared to QIVe for children and high-risk adults in Argentina. The data were robust across the targeted individuals, and the sensitivity analyses were consistent with the primary analysis.

We found that vaccination with QIVc would reduce the number of seasonal influenza cases, complications, hospitalizations, and deaths compared to QIVe. Endorsing this evidence, several studies evaluated the cost-effectiveness of QIVc compared to QIVe worldwide and demonstrated similar findings [17,18,19,21]. This result can be attributed to the fact that cell-based influenza vaccines can be more effective than traditional egg-based influenza vaccines, favoring several populations and clinical and economic outcomes [17].

Egg manufacturing processes have been the primary way of producing influenza vaccines for decades owing to the time-tested methods and established infrastructures. However, one of the primary drawbacks of egg-based vaccines is the egg-adaptive mutation, which alters antigenicity and decreases vaccine immunogenicity and effectiveness [12,47,48]. Cell-based vaccines could provide improved health benefits and cost-savings during flu seasons when egg-adaptation issues are more prevalent. The current analysis showed that in seasons when egg-adaptive mutations predominate, QIVc vaccination compared to QIVe would show its higher clinical preventive benefits and the highest economic savings.

The strengths of this study include the use of a simple and transparent model structure. Both deterministic and probabilistic sensitivity analyses were performed to examine the uncertainty in model parameters, indicating the most sensitive parameters on the model result and proving the model robustness.

This study still has some limitations. Firstly, the lack of local data regarding the probability of influenza complications and health utilities my lead to some bias in the model’s results; however, the latter should impact similarly to both influenza quadrivalent vaccines analyzed. Secondly, the analysis conducted by this study was performed using a pre-pandemic scenario, not involving data related to COVID-19. Therefore, the results in a co-circulation scenario may significantly vary our results, especially with the use of masks and social distancing policies, which have an impact of a reduced number of influenza cases. Thirdly, there have been a limited number of years to demonstrate the effectiveness of QIVc (3 years) and more seasons are needed to validate our findings in a pediatric and adult population. Fourthly, some authors believe that thresholds based on national-income-based approaches have major weaknesses and suggest developing a new framework for articulating cost-effectiveness in a global health policy, considering local policy, programs options, and funding sources [49]. However, in the absence of official data, we followed the same methodology as previous Argentinean cost-effectiveness analyses [4]. Finally, a one-year period was used in this analysis; therefore, the model only calculates the short-term costs and the clinical impact of vaccination programs for a single average influenza season. Thus, the present analysis could be considered as conservative because it does not include influenza-associated sequelae prevention or other long-term effects.

## 5. Conclusions

This economic analysis in high-risk populations (aged < 65 years old) in Argentina indicates that the higher effectiveness of QIVc compared to QIVe could significantly reduce the number of seasonal influenza cases, complications, hospitalizations, and deaths, demonstrating that the implementation of this alternative would be a cost-effective strategy. In addition, QIVc vaccination is expected to be cost-saving over QIVe considering the societal perspective. The clinical and economic benefits of QIVc use would be even higher in seasons with high egg-adaptation issues.

## Figures and Tables

**Figure 1 vaccines-10-01627-f001:**
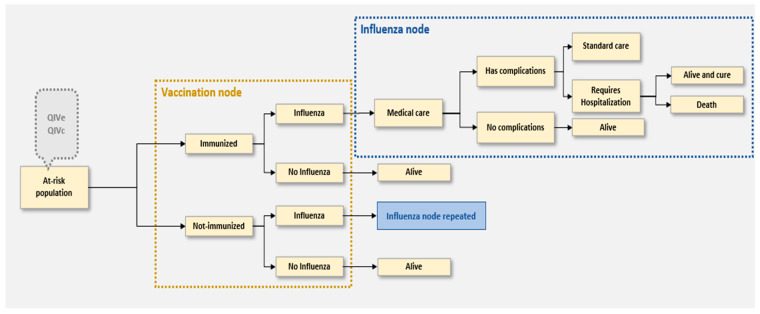
Decision tree model structure.

**Figure 2 vaccines-10-01627-f002:**
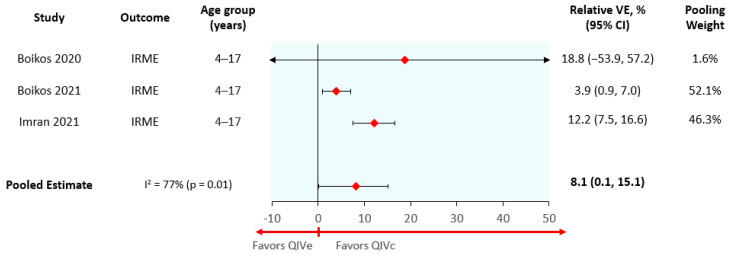
Meta-analysis of QIVc vs. QIVe, prevention of influenza related medical encounters in pediatric population. IRME: influenza-related medical encounters; VE: Vaccine Effectiveness; QIVe: Egg-based Quadrivalent Influenza Vaccine; QIVc: Cell-based Quadrivalent Influenza Vaccine.

**Figure 3 vaccines-10-01627-f003:**
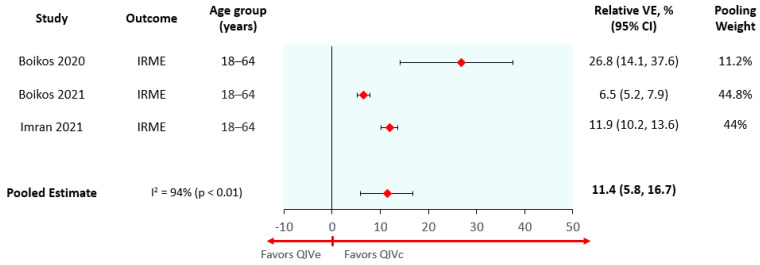
Meta-analysis of QIVc vs. QIVe, prevention of influenza related medical encounters in adult population.IRME: influenza-related medical encounters; VE: Vaccine Effectiveness; QIVe: Egg-based Quadrivalent Influenza Vaccine; QIVc: Cell-based Quadrivalent Influenza Vaccine.

**Figure 4 vaccines-10-01627-f004:**
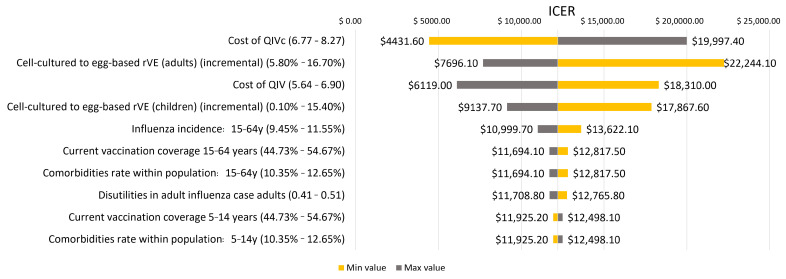
When assessing the results under WHO willingness-to-pay threshold recommendations and using the payer’s perspective, QIVc had a 39.0% chance of being highly cost-effective and 65.1% of being cost-effective when compared to QIVe vaccination [41]. QIVc: Quadrivalent Influenza vaccine—cell-based; QIVe: Quadrivalent Influenza Vaccine—based; rVE: relative Vaccine Effectiveness; y: years; ICER: Incremental Cost-Effectiveness Ratio.

**Figure 5 vaccines-10-01627-f005:**
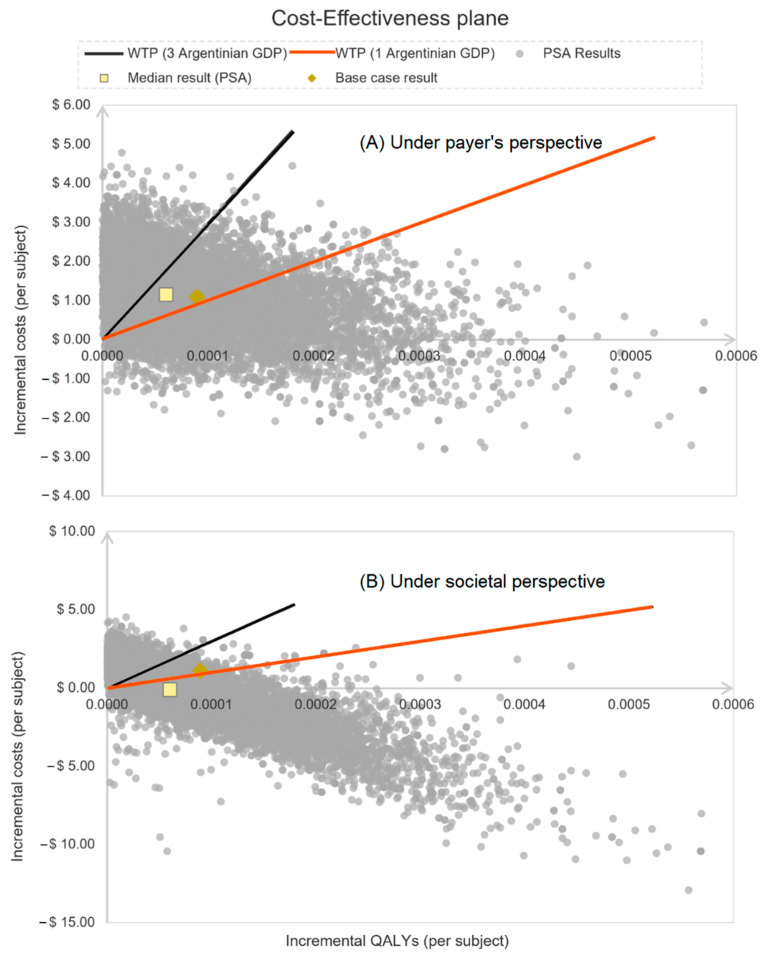
Cost-effectiveness plane of QIVc vs. QIVe comparison.

**Figure 6 vaccines-10-01627-f006:**
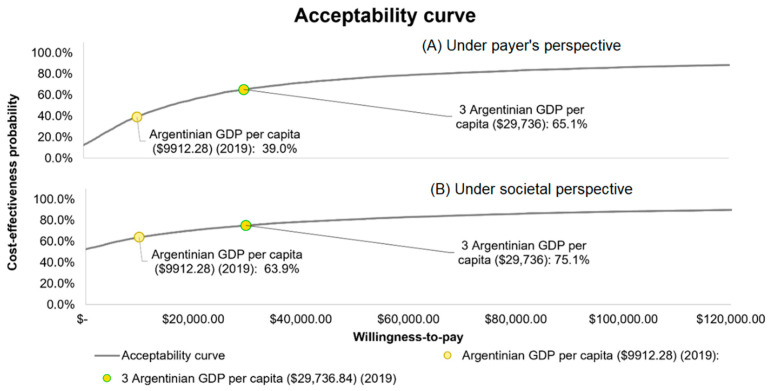
Cost-effectiveness acceptability curve of QIVc vs. QIVe.

**Table 3 vaccines-10-01627-t003:** Model results—base case.

Outcome	QIVc	QIVe	Difference (QIVc − QIVe)
Number of GP visits ^†^	312,010	329,552	−17,541
Number of complications *^†^	43,478	45,896	−2418
Number of influenza-related hospitalizations	6349	6664	−316
Number of influenza-related deaths	248	261	−12
QALYs	33,357,208	33,356,927	282
Vaccination costs	USD 23,746,747.82	USD 18,884,851.69	USD 4,861,896.14
Medical care costs	USD 27,272,877.34	USD 28,696,051.82	USD −1,423,174.48
Absenteeism costs	USD 38,060,335.60	USD 40,198,038.48	USD −2,137,702.89
Early death costs **	USD 58,763,713.66	USD 61,618,116.82	USD −2,854,403.16
Total direct costs	USD 51,019,625.16	USD 47,580,903.51	USD 3,438,721.65
Total costs (both direct and indirect)	USD 147,843,674.42	USD 149,397,058.81	USD −1,553,384.39

* Complications include acute otitis media, respiratory complications such as bronchiolitis and pneumonia, and non-respiratory complications such as cardiovascular, kidney, or neurologic disease. ** Estimated with yearly GDP per capita throughout economically active age of the death subject. An annual 3% discount rate was applied. ^†^ Outcomes estimated through incidence and complication rates for both vaccines. QALY: Quality Adjusted Life Years; QIVe: Quadrivalent Influenza Vaccine–egg-based. QIVc: Quadrivalent Influenza Vaccine–cell-based; GP visits: General Practitioner visits.

## Data Availability

Not applicable.

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
