# Peer review of "Cost-Effectiveness Analysis of Cell Versus Egg-Based Seasonal Influenza Vaccination in Children and Adults in Argentina"

_vaccines, 2022, doi:10.3390/vaccines10101627_

Round 1
Reviewer 1 Report
The manuscript is well written and addresses a relevant problem, as the cost-effectiveness analysis of a new influenza vaccination strategy in Argentina. The analysis is performed adequately, although it considers the direct effects only of vaccination (thus it is conservative) and many parameters (such as the complication rates) are taken from the international literature for lack of local data.
I list below some points in which more details should be provided, or the presentation should be improved.
- in the sentence at line 66, I believe that the authors mean that “…the effectiveness of influenza vaccines can be increases as much as 9%...”;
- the assumption (line 158) that only hospitalized individuals could progress to death should be justified. Several studies (e.g. Rosano et al, 2019), analyzing excess mortality, show that impact of influenza on deaths is higher than what can be gathered from hospitalization data.
- indirect costs from deaths (lines 181-183): the text correctly states that they depend on the number of years between the subjects death age (not considering, I suppose, years before working age for young patients) and retirement age. However, Table 2 shows a single numeric value, without considering age; even if the value is based on the mean age of the dead, this may change with vaccination, I suppose. Furthermore, the authors do not mention any discount rate in this cost; is it neglected?
- Section 2.4. I am not convinced by a specific consideration of season with higher rVE. After all, the sensitivity to that parameter is already shown in Figure 4. If the authors wish to consider specifically the estimate for the season 2017-18 (that has a very high uncertainty), they should also consider the estimate for season 2018-19, in order to provide an overview of how the impact can change from one season to another.
- A few observations about Figure 4. First of all, it would be nice if the authors could increase the font. Moreover, the legend to that Figure seems not correct. Finally, it is stated that the ranges for each parameter are taken from the 95% CI; however, from Table 1 this is not clear for the proportion of comorbidities in the population, and for the influenza incidence .
- Finally, in the text, I found in a few occasions “[Error! Reference source not found]”. Definitely something is wrong.
Aldo Rosano, Antonino Bella, Francesco Gesualdo, Anna Acampora, Patrizio Pezzotti, Stefano Marchetti, Walter Ricciardi, Caterina Rizzo,
Investigating the impact of influenza on excess mortality in all ages in Italy during recent seasons (2013/14–2016/17 seasons),
International Journal of Infectious Diseases,
Volume 88,
2019,
Pages 127-134,
ISSN 1201-9712,
https://doi.org/10.1016/j.ijid.2019.08.003
Author Response
We thank the Editor and Reviewers for the accurate revision provided to our article. We tried to address all of them and are glad to resubmit with the suggested changes. All answers and modifications are highlighted in the main text.
- In the sentence at line 66, I believe that the authors mean that“…the effectiveness of influenza vaccines can be increases as much as 9%...”;
Thank you for the kind suggestion the suggested change was added to the text in line 66.
- the assumption (line 158) that only hospitalized individuals could progress to death should be justified. Several studies (e.g.Rosano et al, 2019), analyzing excess mortality, show that impact of influenza on deaths is higher than what can be gathered from hospitalization data.
We thank you for the recommendation. A more comprehensive description and reasons for this rationale was described in lines 158-162 to increase the comprehension.
- indirect costs from deaths (lines 181-183): the text correctly states that they depend on the number of years between the subjects death age (not considering, I suppose, years before working age for young patients) and retirement age. However, Table 2 shows a single numeric value, without considering age; even if the value is based on the mean age of the dead, this may change with vaccination, I suppose. Furthermore, the authors do not mention any discount rate in this cost; is it neglected?
Thank you for this suggestion. We believe that the description at table 2 was inaccurate indeed. That value actually means the yearly impact of an early death - that value is projected over the number of years in which the dead subject would be at economically active age (from 18 - 65 years old). The authors had corrected those amounts into current values following the Argentinean pharmacoeconomic guidelines, with a 3% annually discount rate. (https://www.iecs.org.ar/en/health-technology-assessment-and-health-economics/economic-evaluations/).
- Section 2.4. I am not convinced by a specific consideration of season with higher rVE. After all, the sensitivity to that parameter is already shown in Figure 4. If the authors wish to consider specifically the estimate for the season 2017-18 (that has a very high uncertainty), they should also consider the estimate forseason 2018-19, in order to provide an overview of how the impact can change from one season to another.
Thank you for your question. The main objective of this section is to bring focus to the benefits of vaccination with QIVc in seasons with high egg-adaptation of influenza circulant viruses, such as those that occurred in 2017-18. The other seasons are not of major interest since they do not present high egg-adaptation and could be well represented by the base case scenario. An improved description for this rationale was added in the lines 210-213.
- A few observations about Figure 4. First of all, it would be nice if the authors could increase the font. Moreover, the legend to that Figure seems not correct. Finally, it is stated that the ranges for each parameter are taken from the 95% CI; however, from Table 1 this is not clear for the proportion of comorbidities in the population, and for the influenza incidence.
Thank you for this highlight. The image 4 was changed to better visualization. We adjusted the table 1 to present the proportion for both the parameters now.
- Finally, in the text, I found in a few occasions “[Error! Referencesource not found]”. Definitely something is wrong.
Thank you for this input. We tried to find the error but we could not. Despite that we revised the new submitted document to make sure all is correct. Thank you again.
We thank all the referees for the revision, and we hope that with this new and improved version of our article can now be suitable to be published by Vaccines. We are at your disposal for any other enlightenment if necessary.
With kind regards,
Analia Urueña
Corresponding Author

Reviewer 2 Report
This study assessed the cost-effectiveness of QIVc. The results suggested that compared to QIVe, QIVc is cost-effective from both the payer and the societal's perspectives. Overall, the manuscript is well organized and the methods are sound. Specific comments are provided below:
1. Section 2.2, line 117-118. While the aothors used 2014-2019 data to estimate influenza curculating strains, the uncertainity from the different proportion of influenza circulating strains should be further assessed by sensitivity analysis.
2. Section 2.2, line 127-130. Same as above, as the relative effectiveness between QIVc and QIVe is a key factor that affects the result (as indicated in the one-way SA results as well). Sine the relative effectiveness of QIV came from a non-systematic review, this parameter should be further assessed and discussed in the study. Maybe the authors could assess at what relative effectiveness level QIVc will become not cost-effective.
3. Table 1. Base case number of GP visits and number of complications were not provided, but these two types of resource utilization were reported in the results.
4. Table 1. not sure why the authors put 2 years old patients into the 2-4 years old group, not the 6-23 months group. It seems more reasonable to put 2-year old patients into the 6-24 month-old group since the Argentina immunization program covers children 6 months to 2 years. A relevant question is why the vaccine coverage rate is high (83%) in Table 1 as the immunization program does not cover children >2 without comorbidities.
5. Table 1. Following the question above, I'm not sure whether probabilities of complications or hospitalization in subjects without complications apply to patients aged > 2. Again, as the immunization program only covers patients with comorbidities or complications aged between 2 to 64 years old.
6. Results from Figure 4-6 should be reported or summarized in the text as well.
Author Response
We thank the Editor and Reviewers for the accurate revision provided to our article. We tried to address all of them and are glad to resubmit with the suggested changes. All answers and modifications are highlighted in the main text.
- Section 2.2, line 117-118. While the aothors used 2014-2019 data to estimate influenza curculating strains, the uncertainity from the different proportion of influenza circulating strains should be further assessed by sensitivity analysis.
Thank you for your suggestion. This parameter was also assessed in the sensitivity analysis (both deterministic and probabilistic). However, it was not found as the most relevant (as per tornado diagram in figure 4). We added a sentence in the sensitivity analysis section to improve the description of such analysis.
- Section 2.2, line 127-130. Same as above, as the relative effectiveness between QIVc and QIVe is a key factor that affects the result (as indicated in the one-way SA results as well). Sine the relative effectiveness of QIV came from a non-systematic review, this parameter should be further assessed and discussed in the study. Maybe the authors could assess at what relative effectiveness level QIVc will become not cost-effective.
Thank you for your highlight. This parameter was also assessed in the sensitivity analysis (both deterministic and probabilistic). Added a phrase discussing the cut-off point for cost-effectiveness, as requested. We added in lines 127-130 (“The relative vaccine effectiveness (rVE) of QIVc vs. QIVe was estimated using a non-systematic structured review to identify publications reporting relative vaccine effectiveness estimates for interventions and comparators for prevention of influenza-related medical encounters (IRME) in any clinical setting”)]
- Table 1. Base case number of GP visits and number of complications were not provided, but these two types of resource utilization were reported in the results.
Thank you for your note. These outcomes were estimated from incidence and complication rates, both reported in table 1. Added a footnote clarifying this point in table 3.
- Table 1. not sure why the authors put 2 years old patients into the 2-4 years old group, not the 6-23 months group. It seems more reasonable to put 2-year old patients into the 6-24 month-old group since the Argentina immunization program covers children 6months to 2 years. A relevant question is why the vaccine coverage rate is high (83%) in Table 1 as the immunization program does not cover children >2 without comorbidities.
Thank your explanation. 83% coverage is for high-risk children between 2 and 4 years old. This was described in paragraph 2.2: “For the groups between 2 and 64 years old, only the population with risk factors was considered in the model, with estimates provided by the National Immunization Program (NIP) and based on the National Risk Factors Survey [28]. The vaccination coverage rates for each age group were provided by the NIP and were used in the model for both vaccinations”.
- Table 1. Following the question above, I'm not sure whether probabilities of complications or hospitalization in subjects without complications apply to patients aged > 2. Again, as the immunization program only covers patients with comorbidities or complications aged between 2 to 64 years old.
Dear Reviewer, Thank you for pointing this out. You’ve noticed correctly: as the model embraces the vaccination of only at-risk (with comorbidities) subjects older than 2 years old, the description of probabilities for complications and hospitalization in individuals older than 2 years old without such characteristics makes no sense. To improve understanding, we removed the probabilities for such parameters in table 1.
- Results from Figure 4-6 should be reported or summarized in the text as well.
Thank you for noticing this. We added in lines 267-274 a new paragraph with an improved the description of the images. And text added with explanation.
“The probabilistic sensitivity analysis for the cost-effectiveness under the payer perspective confirmed the base case scenario, where QIVc was more effective in reducing influenza cases and complications, but also showed higher overall costs than QIVe (Figure 5). When assessing the results under WHO willingness-to-pay threshold recommendations and using the payer’s perspective, QIVc had a 39.0% chance of being highly cost-effective and 65.1% of being cost-effective when compared to QIVe vaccination [42] (Figure 6). QIVc would not be considered cost-effective if its relative effectiveness over QIVe was decreased as low as 3.69% for adults or -7% for children.”
We thank all the referees for the revision, and we hope that with this new and improved version of our article can now be suitable to be published by Vaccines. We are at your disposal for any other enlightenment if necessary.
With kind regards,
Analia Urueña
Corresponding Author
